# Exploring the recurrent states of football teams' tactical organization on the pitch during Brazilian official matches

Felipe Arruda Moura[1,2©¤]*, Murilo José de Oliveira Bueno[1©], Fabio Giuliano Caetano[1©], Maisa Silva[3©], Sergio Augusto Cunha[4©], Ricardo da Silva Torres[2,5©]*

1 Laboratory of Applied Biomechanics, Sport Sciences Department, State University of Londrina, Londrina, Brazil, 2 Wageningen Data Competence Center, Wageningen University and Research, Wageningen, The Netherlands, 3 Institute of Computing, University of Campinas, Campinas, Brazil, 4 College of Physical Education, University of Campinas, Campinas, Brazil, 5 Department of ICT and Natural Sciences, NTNU-Norwegian University of Science and Technology, Ålesund, Norway

© These authors contributed equally to this work.
¤ Current address: Sport Sciences Department, State University of Londrina, Londrina, Brazil
* ricardo.torres@ntnu.no (RST); felipemoura@uel.br (FAM)

**Data Availability Statement:** All data supporting the findings of this study are available within the paper and its Supplementary Information. The primary data of the Multiscale Fractal Dimension

## Abstract

Football teams' tactical organization on the pitch is usually represented by the surface area. Considering the different shapes adopted by the teams during the match, the role of the tactical variability for success is lacking. The aim of this study was to explore and to evaluate the association between recurrent states of tactical organization and technical performance during football matches. A total of 28 teams of Brazilian First Division Championships were analysed. Teams' surface area shapes were represented by the maximum value of the Multiscale Fractal Dimension in each timestamp, producing a time series. Recurrences of states of tactical organization were determined via recurrence plots and recurrence quantitative analysis during attacking and defending phases, and considering the whole match. The outcomes were correlated with nine traditional technical performance indicators. The main results showed that structural recurrence or variability on tactical organization is associated with performance success during the defending and attacking actions. Recurrence plot and measures based on the recurrence density proved to be valuable tools to represent teams' dynamics.

## Introduction

Football is a complex system with different levels of organization, represented by the teams, by groups of players, and by each player itself, from a macro, meso, and micro perspective, respectively. This dynamical system has an auto-organization that emerges from interactions among its elements, so individual behavior reflects on group/team behavior [1, 2]. From players' spatiotemporal raw data, macro structure behaviour is represented by aggregated features, such as team centroid, team spread, and team surface area [3–5].

curves of the teams' tactical organization in each timestamp are available at: \url{https://doi.org/10.6084/m9.figshare.24494983.v1}.

**Funding:** This work was supported by the Brazilian National Council for Scientific and Technological Development (CNPq) under grants \#401004/2022-8, \#200290/2022-3, and \#305997/2022-0. The funders had no role in study design, data collection and analysis, decision to publish, or preparation of the manuscript.

**Competing interests:** The authors have declared that no competing interests exist.

Usually, team surface area is defined as the area of the convex hull defined by outfield team-mates players as a function of time [6]. Surface area is an important tactical performance indicator that is associated with success in attacking and defending actions [4, 7, 8]. Recently, a systematic review provided reference values of tactical variables from male professional matches, including teams' surface area, with values ranging from 750 to 1,831 $m^2$ considering Brazilian, Portuguese, English, and Spanish leagues [5]. However, considering that different structures of the players on the pitch may provide similar surface areas or, for similar organization structures, it is possible that the teams present different areas and shapes [9]. Recently, Bueno and colleagues [9] validated the Multiscale Fractal Dimension [10] as an accurate shape descriptor of convex hull polygon, advancing in the tactical description of the teams beyond the representation only by the magnitude of the area. With a set of shape descriptors, authors were able to identify the relative frequency of clusters of structures throughout the match, encoding therefore how variable collective tactical behavior can be [9].

During a football match, when a team has possession of the ball, their objective is to maintain control and score a goal, while the opposing team simultaneously strives to regain possession and prevent the team in control from scoring. As a result, teams establish well-coordinated behaviors, leading to stable player patterns on the field [11]. These connections between players are variable and nonlinear in nature, as an individual player's mistake can either have a negligible impact or a significant one [12].

From a dynamical systems perspective, the role of the variability has been investigated in several research areas, from human movement analysis [13] to collective behaviour in sports approaches [4, 14–17]. Variability analysis can be defined as the evaluation of the degree and patterns of variation over specific time intervals [18]. Since the development of players tracking technologies, multiple methods of variability analysis have tried to characterize how athletes from different sports behave during competition, and how variability changes as a response to specific stimuli, such as entropy measures [19], vector coding analysis [11], spectral analysis [15, 16], among others. In football match analysis, the variability of players' distribution on the pitch (i.e., changes in inter-team coordination) has been associated with successful attacks [11]. Also, a previous study [4] discussed the role of the players' positional variability (extracted from Principal Component Analysis) of the top-ranked teams during Euro 2012. Variability was also explored to determine team formation and player role change-point based on the sequence of role-adjacency matrices. The method allowed, for instance, to discover the switching patterns or to detect set-pieces patterns [20]. Recently, a method to describe team formation was also proposed to discover team structures automatically from players tracking data, by minimising the entropy of a set of player role distributions, separating the player tracking data into distinct role distributions to allow the discovery of the underlying team structure [21]. The method provided insights into formation visualisation, formation clustering, and role-based player analysis.

Several research fields provide a set of tools to analyse behaviour of complex systems, and some of them were applied to football match analysis. For instance, the T-patterns approach was used to detect and describe recurring sequences of events (such as pass, tackle, header, run, etc) in specific zones of the football pitch [22]. The authors showed that the number of patterns found is highly correlated with an assessment of the team's performance by experts. A non-linear time series analysis tool for visualizing recurring patterns of complex and dynamical system behaviour is the Recurrent Plots and the recurrence quantification analyses that allows quantifying properties of the recurrent states [23]. In football, a previous study [12] aggregated all players' 2D coordinates in each timestamp position into the mean positional change of all players to assess the similarity between the two states. The authors showed that the technical performance during the matches (i.e., traditional game-related statistics as the

number of shots to goal, goals scored, correct passes, corners, etc) are associated with recurring states during the match. A similar approach [24] calculated the average distance of all the players between their positions for each pair of seconds and used Recurrence Plots to analyze players' positional dynamics. The recurrence parameters showed several significant correlations with traditional performance indicators like the number of goals and passes completed.

The studies described above presented important highlights about the complexity of team behaviour and manners to objectively evaluate it in the light of dynamical systems tools. However, one can argue that the identification of recurrent states of tactical features, such as the shape of team organization on the pitch, may elucidate the role of team behavior variability as a key performance indicator. In other words, to identify the relation between the team's technical performance and a more (or less) structured manner in which a team is organized is relevant to providing insights about the pros and cons of the predictability of tactical actions.

Therefore, the purpose of this study was to explore and evaluate the association between the recurrent states of surface area shapes and technical performance during the attacking and defending phases of professional football matches. For football analysis, the technical performance represented by the game-related statistics may provide valuable information about team performance and can discriminate between successful and unsuccessful teams [25–28]. We hypothesized that a) during defending actions, a more recurrent shape will be associated with success in technical performance and that b) the more variable the structure of the team shape is during ball possession, the higher the success in attacking actions.

## Materials and methods

### Study design

In the present study, the recurrence states of the tactical organization on the pitch were analyzed in official matches. Players' trajectories were determined using a video-based tracking system. From players' positions, the shapes of the team surface area on the pitch were described using the maximum value of the Multiscale Fractal Dimension throughout the matches. Then, recurrence quantification analyses of recurrence plots from the Multiscale Fractal Dimension were calculated and the outcomes were associated with technical performance indicators (passes, shots on goal, goals scored, ball possession, etc).

### Participants

This study used retrospective datasets freely provided by Bueno and colleagues [9]. A total of 28 teams during 14 official matches of the Brazilian First Division Championships were analyzed. All the matches were complete and with no red cards or injuries that may have resulted in long periods with less than 10 outfield players for each team. The 2D trajectories of 366 players were obtained via DVideo software [29, 30]. DVideo software allows automatic player tracking from video sequences of up to six cameras fixed at high places of the stadiums, using image segmentation algorithms. Data were filtered using a 3rd-order zero-lag low-pass Butterworth filter with cutoff parameters defined by spectral and residual analysis. This study was conducted in agreement with the ethical recommendations of the Declaration of Helsinki, with informed consent obtained for players' data collection. The study was approved by the Research Ethics Committee of the State University of Londrina (#6.495.956). The names of the teams and players involved were kept confidential.

The DVideo software was used to manual register the technical actions of the players during the football matches. The software enables the identification of various actions, such as dribbling, passing, controlling, shooting at the goal, and tackling, as well as notable events like corner kicks, fouls, goals, goal kicks, offsides, and throw-ins. The system also allows for

identifying whether each action was successful or unsuccessful, along with the player responsible for executing it. The reliability of this notational analysis procedure has an intra-rater agreement of 97.8% and an inter-rater agreement of 93.9% [31].

## Team organization and Multiscale Fractal Dimension

Teams' tactical organizations on the pitch were described by the convex hull defined by out-field teammates players at each timestamp [4]. Then, binary images of the convex polygon of each team separately were created and stored for further analysis using Multiscale Fractal Dimension.

Fractal Dimension is largely used to describe object shape complexity, texture, and geometric composition [32]. In the present study, the Minkowski-Bouligand fractal dimension was used to describe teams' shape as a function of different dilations (scales), a method referred to in the literature as Multiscale Fractal Dimension (for a detailed description, see [9]). To calculate the Minkowski-Bouligand fractal dimension for each polygon, the algorithm consists of three steps: 1) Computing the Euclidean Distance Transform (EDT), using the polygon contour as input; 2) Evaluating the areas of dilated contours defined in terms of a radius $r$, considering that each multiscale instance $S(r)$ of the original form $S$ is obtained by the threshold of the cost map computed by the EDT and calculating $A(r)$ as the area of the respective dilated version of the shape through the cumulative histogram of the cost map; 3) Estimating of the multiscale fractal dimension, defined as the following equation (Eq 1):

$$MultiscaleFractalDimension = 2 - \lim_{x \to 0} \frac{log(A(r))}{log(r)} \tag{1}$$

For each timestamp, the multiscale fractal dimension curve was generated and the maximum value was identified and registered over time as a proxy of the team's tactical organization. Fig 1 represents the method workflow, from data collection to recurrence plot computation.

## Recurrence plots

The maximum value of the multiscale fractal dimension was compiled for all polygons over time, producing a time series. As our focus is on recurrences of states of a dynamical system represented by the shape of each team, recurrences of a trajectory $\vec{x}_i \in \mathbb{R}^d$ in phase space were measured, i.e. the recurrence plot. The recurrence plot allows the visualization of the recurrent states that meet a distance-based threshold criterion, formally expressed by the matrix [23]:

$$R_{i,j}(\varepsilon) = \Theta(\varepsilon - \|x_i - x_j\|), \qquad i,j = 1, \cdots, N. \tag{2}$$

where $N$ is the number of measured points $\vec{x}_i$, $\varepsilon$ is a threshold distance, $\Theta$ the Heaviside function (i.e., $\Theta(x) = 0$, if $x < 0$, and $\Theta(x) = 1$, otherwise) and $\|\|$ is a norm (e.g., Manhattan). In the present study, $\varepsilon$ was set as 0.01 considering an approximate distance between different clusters of fractal dimension curves that represent football teams shapes [9].

For $\varepsilon$-recurrent states, i.e., for states which are in an $\varepsilon$-neighbourhood, we defined $R_{i,j} \equiv 0$. The recurrence plot was obtained by plotting the recurrence matrix, Eq 2, and using different colours for its binary entries. Both axes of the RP are time axes. Time is represented in the recurrence plot matrix rightwards and upwards (convention). Because of the interest in the evaluation of the recurrence plots considering the distinction between the attacking (when the team, both in timestamps $i$ and $j$, was with ball possession) and defending phases (when the team, both in timestamps $i$ and $j$, was without ball possession), separate recurrence plots were

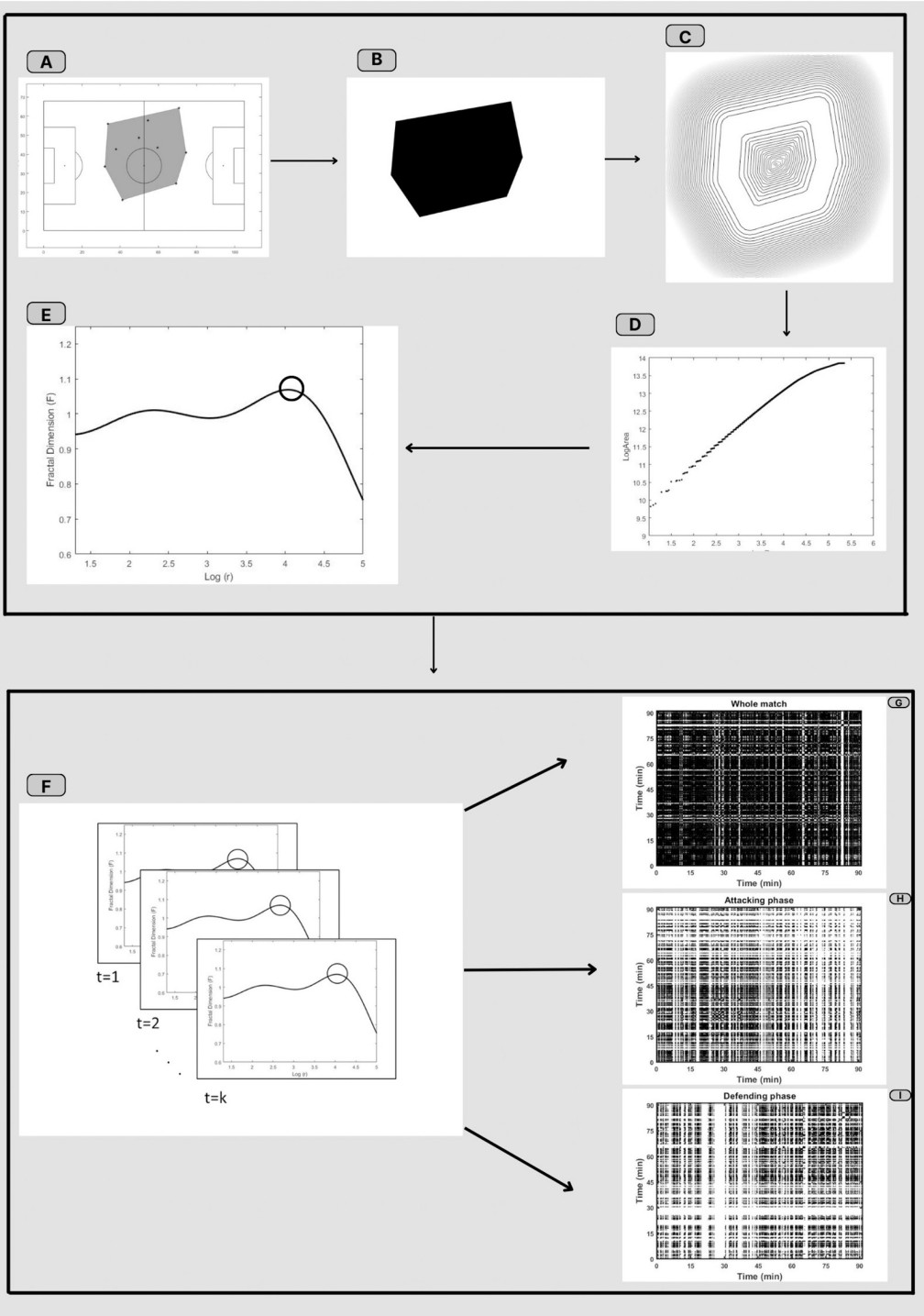

**Fig 1. The methodology workflow for computing the recurrence plot computation.** From the players' position on the pitch, the convex hull was extracted (A), and then a binary image of the shape was stored (B). From the dilated contours of the shape (C), the Logarithmic function of the cost (areas of the dilated contours) as a function of the dilation radius (D) was determined to calculate the Multiscale Fractal Dimension curve (E). The maximum fractal in each timestamp was stored as a time series from the $k$ timestamps (F) to be represented by recurrence plots of the whole match (G), and during the attacking (H) and defending (I) phases.

generated, in addition to the recurrence plot considering the whole game with no phase distinction. For the recurrence plots during the attacking phase, defending phase, and whole match, black dot at the coordinates $(i, j)$, if $R_{i,j} \equiv 1$ was plotted. For all recurrence plots, a white dot was defined when $R_{i,j} \equiv 0$.

## Technical performance indicators

The technical performance during the attacking and defending phases was defined by the nine indicators: tackles performed, correct passes performed, wrong passes performed, opponent wrong passes, shots to goal performed, shots to goal suffered, goals scored, goals suffered, ball possession. Each indicator was defined as follows:

- Tackle: the action of taking the ball from an opponent player.

- Pass: the action of touching the ball aiming to make it reach the teammate. If the ball reached the teammate, it was defined as a correct pass. Otherwise, if the ball was intercepted by an opponent or went beyond the pitch boundaries, it was defined as a wrong pass.

- Shot to the goal: the action of touching the ball clearly aiming to score a goal. In the present study, all the shots to the goal were registered.

- Goal: defined as the event when the ball crossed the goal line and was categorized into goal scored and goal suffered.

- Ball possession: it was considered that a team recovered ball possession when two consecutive technical actions were performed by a given player (i.e., two actions performed by a single player or one action performed by two different players on the same team).

All the indicators were defined as absolute frequencies, except the ball possession, represented as relative frequency.

## Statistical analysis

Measures of complexity that quantify the small-scale structures in each recurrence plot, named recurrence quantification analysis [23], were extracted, as follows:

- Recurrence rate: represents a measure of the density of recurrence points in the RP, defined as Eq 3:

$$RR(\varepsilon) = \frac{1}{N^2} \sum_{i,j=1}^{N} R_{i,j}(\varepsilon)$$

(3)

- Determinism: defined as the ratio of recurrence points that form diagonal structures (of at least length $l_{min} = 8$ s) to all recurrence points, represents a measure for predictability of the system, as Eq 4:

$$DET(\varepsilon) = \frac{\sum_{l=l_{min}}^{N} lP(\varepsilon, l)}{\sum_{l=1}^{N} lP(\varepsilon, l)}$$

(4)

where $P(\varepsilon, l)$ is the histogram of diagonal lines of length $l$.

- Diagonal line length: a diagonal line means that a segment of the time series is rather close during $l$ time steps to another segment of the time series at a different moment. Two measures are based on diagonal line length, the average diagonal line length ($L$) and the longest

diagonal line found in the recurrence plot ($L_{max}$), defined as Eqs 5 and 6.

$$L(\varepsilon) = \frac{\sum_{l=l_{min}}^{N} lP(\varepsilon, l)}{\sum_{l=l_{min}}^{N} P(\varepsilon, l)} \tag{5}$$

$$L_{max}(\varepsilon) = max(l_{ii=1}^{N_l}) \tag{6}$$

where $N_l$ is the number of diagonal lines.

- Entropy: reflects the complexity of the recurrence plot in respect of the diagonal lines (i.e., for uncorrelated noise, the value of entropy is small, indicating its low complexity), and refers to the Shannon entropy of the probability $p(l) = P(\varepsilon, l)/N_l$ to find a diagonal line of exactly length $l$ in the recurrence plot, as Eq 7.

$$ENTR(\varepsilon) = -\sum_{l=l_{min}}^{N} p(l) \; ln \; p(l) \tag{7}$$

- Vertical line length: a vertical line marks a time interval in which a state does not change or changes very slowly. The total number of vertical lines of the length $v$ in the recurrence plot is given by the histogram $P(\varepsilon, v)$. Two measures are based on diagonal line length. The laminarity ($LAM$), defined as the ratio between the recurrence points forming the vertical structures and the entire set of recurrence points (Eq 8). The trapping time, defined as the average length of vertical structures (Eq 9).

$$LAM(\varepsilon) = \frac{\sum_{v=v_{min}}^{N} vP(\varepsilon, v)}{\sum_{v=1}^{N} vP(\varepsilon, v)} \tag{8}$$

$$TT(\varepsilon) = \frac{\sum_{v=v_{min}}^{N} vP(\varepsilon, v)}{\sum_{v=1}^{N} P(\varepsilon, v)} \tag{9}$$

where $v_{min} = 8$ s.

Spearman correlation test was used to evaluate the relationship between recurrence quantitative analysis measures and technical performance data. For the attacking phase, the technical performance indicators evaluated were: correct passes performed, wrong passes performed, shots to goal performed, and goals scored. For the defending phase, the following technical performance indicators were evaluated: tackles performed, opponent wrong passes, shots to goal suffered, and goals suffered. When the whole match was considered, all performance indicators were tested. The correlation coefficients were classified (0.0–0.3: Negligible; 0.3–0.5: Low; 0.5–07: Moderate; 0.7–0.9: High; 0.9–1: Very high) as proposed in [33]. A significance level of $p < 0.05$ was used for all statistical analyses.

## Results

Fig 2 presents examples of recurrence plots identified for the team that presented the lowest (A) and highest (B) values of recurrence states when the whole match was considered. Visual inspection of recurrence plots allows to identification of clear patterns during the match. For instance, there are clear moments in which recurrence states of the tactical organization are lacking, around minute 60 for team A during the attacking phase, and between minutes 60–70

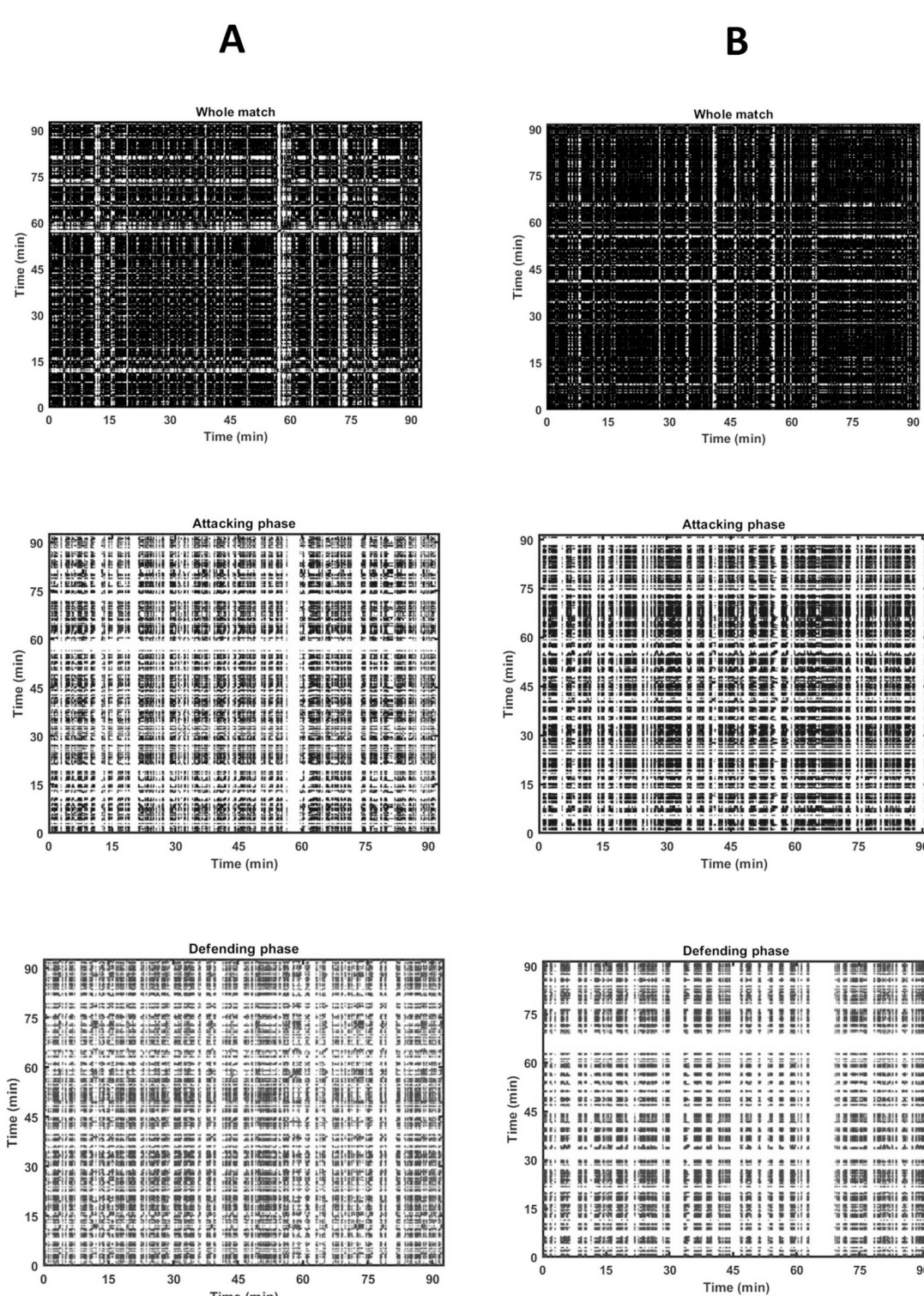

**Fig 2. Recurrence plots of the tactical shapes of the teams that presented the lowest (A) and highest (B) values of recurrence rate when the whole match was selected (in black), and the recurrence plots during the attacking (blue) and defending phases (red).**

during the defending phase of team B. On the other hand, recurrence states are visible between minutes 60–75 during the attacking phase of team B.

Fig 3 presents the violin plots for all recurrence quantitative analysis measures. As expected, the median values are higher for the whole match, when there is no distinction of shape recurrences according to match phases (attacking and defending). Although the shape of the distributions shows values concentrated around the median, for some measures and conditions, it is possible to observe concentrated values on the violin ends (for instance, Fig 3A for attacking and defending phases, Fig 3B–3F for the whole match).

Tables A.3, A.4, and A.5 in S1 Appendix present the outcomes of the recurrence quantitative analysis considering the whole match, the attacking, and defending phases, respectively. Table 1 presents the technical performance of each team analyzed. The outcomes represent the absolute frequency of the events, except for ball possession (relative frequency).

Each recurrence quantitative analysis measure was correlated with team technical performance. Table 2 presents the results of the correlation coefficients considering match phases (whole match, attacking and defending phase). For the whole match, a low positive significant association was found between recurrence rate and correct passes performed ($\rho = .44$). Moderate negative significant association was identified between goals scored with the average diagonal line length, entropy, and trapping time ($\rho = -.54$) and low negative significant association was found between goals scored and determinism ($\rho = -.49$). Low and moderate positive associations were identified between the ball possession with determinism ($\rho = .44$) and laminarity ($\rho = .55$), respectively.

Regarding the attacking phase, a high and low positive significant association was identified between correct passes performed with recurrence rate ($\rho = .72$) and determinism ($\rho = .46$), respectively. Low negative significant association was found between wrong passes performed with determinism ($\rho = -.40$), average diagonal line length ($\rho = -.41$), entropy ($\rho = -.40$) and trapping time ($\rho = -.43$). Additionally, low positive significant association between recurrence rate and shots to goal performed ($\rho = .49$) was found.

Considering the defending phase, a moderate positive significant association was identified between recurrence rate and shots to goal suffered ($\rho = .62$). Moderate and low negative significant associations were found between determinism with tackles performed ($\rho = -.52$) and opponent wrong passes ($\rho = -.38$), respectively. The average diagonal line length and entropy also presented a low negative significant association with tackles performed ($\rho = -.43$ and $\rho = -.45$, respectively).

## Discussion

The main focus of the present study was to explore the variability of surface area shapes of football teams on the pitch from recurrence plots and recurrence quantitative analysis, and the association with the technical performance indicators during the matches. The key results showed that structural recurrence/changes in tactical organization are associated with performance during the defending and attacking phases of the match. To the best of our knowledge, this is the first study with evidence regarding the relationship between tactical organization variability and performance using non-linear techniques from a dynamical system perspective.

To represent teams' structural organization on the pitch, the concept of complexity shape description from Multiscale Fractal Dimension was used. Although other simpler shapes descriptors are available in literature [34], Multiscale Fractal Dimension proved to be valid and more efficient to describe polygons formed by the convex hull of data from the position of football players [9]. From the multiscale fractal curve, the maximum value was identified, which represents the highest shape complexity, and it is invariant to scale.

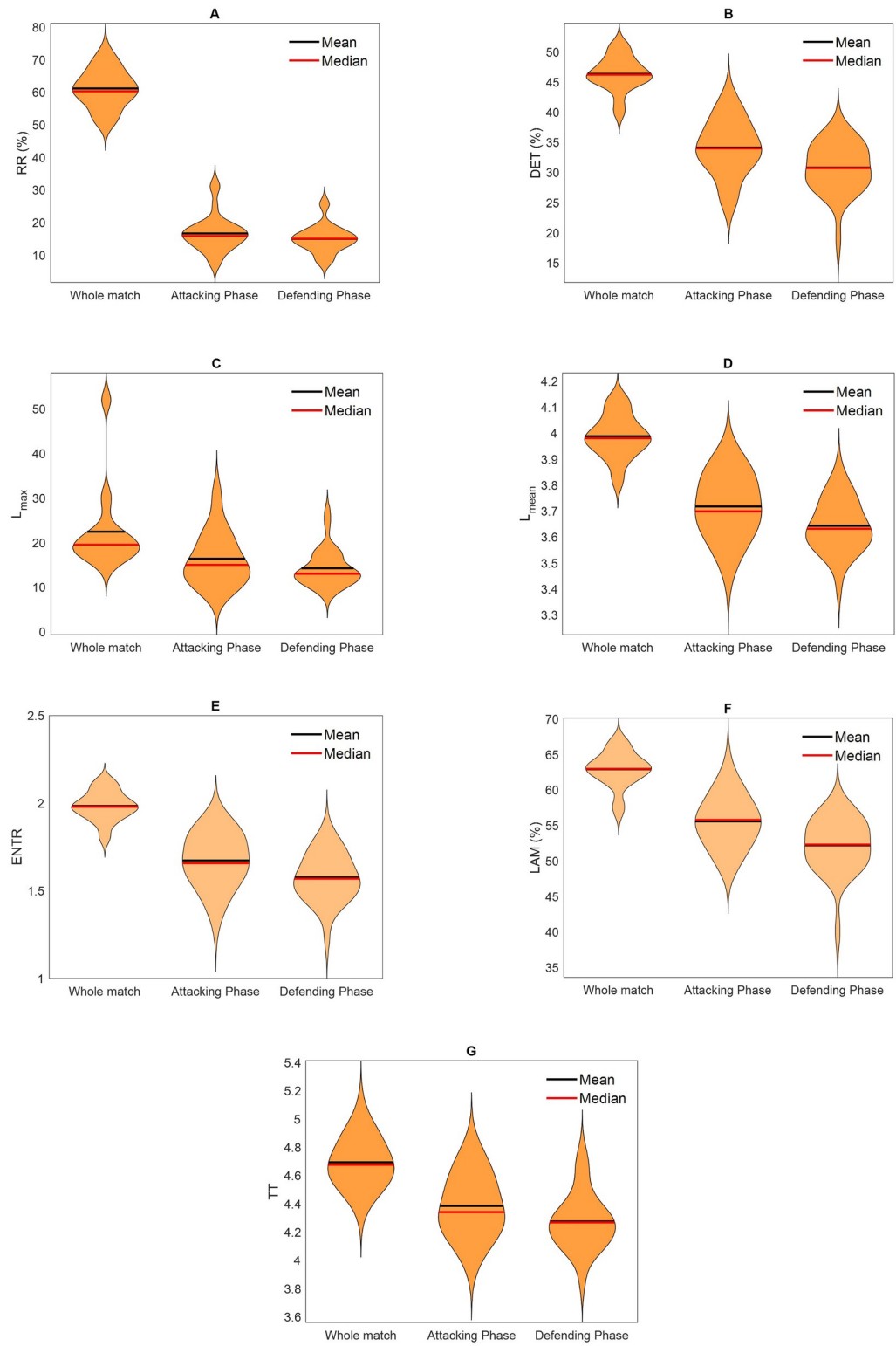

**Fig 3. Recurrence Quantitative Analysis measures (A: Recurrence Rate; B: Determinism; C: Longest diagonal line; D: Average diagonal line length; E: Entropy; F: Laminarity; G: Trapping time) for the whole match, and for the attacking and defending phases.**

**Table 1. Summary of the performance indicators of the teams analyzed.**

| Team | Tackles | correct passes | Wrong passes | Opponent wrong passes | SG | SG suffered | Goals scored | Goals suffered | BP (%) |
|---|---|---|---|---|---|---|---|---|---|
| 1 | 154 | 308 | 118 | 128 | 12 | 21 | 0 | 3 | 44.5 |
| 2 | 155 | 329 | 128 | 118 | 21 | 12 | 3 | 0 | 55.5 |
| 3 | 119 | 187 | 114 | 96 | 13 | 17 | 1 | 1 | 44.7 |
| 4 | 133 | 247 | 96 | 114 | 17 | 13 | 1 | 1 | 55.3 |
| 5 | 177 | 305 | 164 | 148 | 18 | 16 | 1 | 2 | 53.4 |
| 6 | 201 | 226 | 148 | 164 | 16 | 18 | 2 | 1 | 46.6 |
| 7 | 177 | 326 | 135 | 131 | 18 | 15 | 1 | 1 | 51.5 |
| 8 | 154 | 302 | 131 | 135 | 15 | 18 | 1 | 1 | 48.5 |
| 9 | 97 | 339 | 68 | 54 | 20 | 14 | 2 | 1 | 57.6 |
| 10 | 109 | 249 | 54 | 68 | 14 | 20 | 1 | 2 | 42.4 |
| 11 | 89 | 354 | 63 | 76 | 18 | 5 | 1 | 0 | 53.9 |
| 12 | 79 | 246 | 76 | 63 | 5 | 18 | 0 | 1 | 46.1 |
| 13 | 79 | 393 | 84 | 64 | 18 | 12 | 1 | 2 | 55.6 |
| 14 | 102 | 246 | 64 | 84 | 12 | 18 | 2 | 1 | 44.4 |
| 15 | 76 | 423 | 65 | 69 | 12 | 7 | 1 | 3 | 46.6 |
| 16 | 89 | 325 | 69 | 65 | 7 | 12 | 3 | 1 | 53.4 |
| 17 | 105 | 382 | 101 | 106 | 25 | 3 | 0 | 0 | 51.0 |
| 18 | 114 | 292 | 106 | 101 | 3 | 25 | 0 | 0 | 49.0 |
| 19 | 76 | 358 | 81 | 71 | 14 | 8 | 3 | 1 | 48.1 |
| 20 | 92 | 303 | 71 | 81 | 8 | 14 | 1 | 3 | 51.9 |
| 21 | 111 | 414 | 104 | 104 | 21 | 9 | 1 | 0 | 64.9 |
| 22 | 113 | 201 | 104 | 104 | 9 | 21 | 0 | 1 | 35.1 |
| 23 | 119 | 508 | 106 | 113 | 17 | 7 | 1 | 0 | 67.1 |
| 24 | 110 | 245 | 113 | 106 | 7 | 17 | 0 | 1 | 32.9 |
| 25 | 121 | 333 | 102 | 126 | 11 | 9 | 2 | 1 | 50.7 |
| 26 | 121 | 414 | 126 | 102 | 9 | 11 | 1 | 2 | 49.3 |
| 27 | 101 | 353 | 93 | 86 | 12 | 22 | 3 | 1 | 42.3 |
| 28 | 89 | 488 | 86 | 93 | 22 | 12 | 1 | 3 | 57.7 |

SG: Shot to goalBP: Ball possession

The bidimensional representation allowed qualitative analysis of clear patterns of recurrence and/or variability of the tactical structure. Thus, more recurrence points in the figure represent that a given team has similar shapes over time while empty spaces represent the lack of patterns of organization. So, recurrence plots provide important insights into the time evolution of the time series once typical patterns in recurrence plots are linked to specific behaviour of the system represented by the teams' organization. The large-scale patterns in recurrence plots, named typology [35], can be classified as homogeneous, periodic, drift, and disrupted. In Fig 2, it is possible to identify white bands in all match conditions (whole match, attacking, and defending phases), suggesting that the typology can be classified as disrupted. This feature represents abrupt changes in the dynamics, allowing the identification of extreme and rare events easily by using the frequency of their recurrences [23]. When considering the attacking and the defending phases, the white bands may represent the lack of recurrence states or a change in the ball possession. On the other hand, considering the recurrence plots of the whole match, definitively, white bands represent ruptures in the behavior pattern, showing the importance of exploring the recurrence states discriminated by the match circumstances.

**Table 2. Correlation coefficients and significance between performance indicators and quantitative recurrence analysis measures (significant correlations in bold).**

| Whole match | RR | DET | Lmax | L | ENTR | LAM | TT |
|---|---|---|---|---|---|---|---|
| Tackles performed | -.15 (p =.432) | -.01 (p =.953) | .22 (p =.257) | -.04 (p =.825) | -.04 (p =.825) | .00 (p =.991) | -.10 (p =.621) |
| correct passes performed | **.44 (p =.018)** | .27 (p =.164) | -.16 (p =.426) | .18 (p =.360) | .18 (p =.36) | .25 (p =.207) | .13 (p =.508) |
| Wrong passes performed | -.05 (p =.806) | .04 (p =.856) | -.03 (p =.896) | .03 (p =.861) | .03 (p =.861) | -.06 (p =.761) | -.03 (p =.87) |
| Opponent wrong passes | -.04 (p =.857) | .05 (p =.806) | .2 (p =.302) | .09 (p =.657) | .09 (p =.657) | -.02 (p =.907) | .10 (p =.596) |
| Shots to goal performed | .03 (p =.877) | .26 (p =.185) | .00 (p =.981) | .20 (p =.315) | .20 (p =.315) | .33 (p =.089) | .10 (p =.602) |
| Shots to goal suffered | -.27 (p =.162) | -.21 (p =.281) | .08 (p =.697) | -.23 (p =.239) | -.23 (p =.239) | -.29 (p =.140) | -.22 (p =.272) |
| Goals scored | -.26 (p =.186) | **-.49 (p =.009)** | -.26 (p =.174) | **-.54 (p =.003)** | **-.54 (p =.003)** | -.31 (p =.114) | **-.54 (p =.003)** |
| Goals suffered | -.13 (p =.514) | -.28 (p =.148) | .00 (p =.989) | -.34 (p =.075) | -.34 (p =.075) | -.26 (p =.185) | -.22 (p =.263) |
| Ball possession | .31 (p =.109) | **.44 (p =.020)** | .05 (p =.809) | .37 (p =.054) | .37 (p =.054) | **.55 (p =.002)** | .16 (p =.403) |
| **Attacking phase** | RR | DET | Lmax | L | ENTR | LAM | TT |
| correct passes performed | **.72 (p =.000)** | **.46 (p =.013)** | -.05 (p =.798) | .31 (p =.114) | .30 (p =.123) | .32 (p =.092) | .19 (p =.338) |
| Wrong passes performed | -.11 (p =.594) | **-.40 (p =.033)** | -.28 (p =.149) | **-.41 (p =.032)** | **-.40 (p =.036)** | -.26 (p =.175) | **-.43 (p =.021)** |
| Shots to goal performed | **.49 (p =.008)** | -.04 (p =.858) | .00 (p =.989) | -.14 (p =.491) | -.15 (p =.461) | -.07 (p =.707) | -.30 (p =.124) |
| Goals scored | .14 (p =.489) | -.16 (p =.406) | .03 (p =.868) | -.27 (p =.161) | -.27 (p =.161) | -.17 (p =.377) | -.27 (p =.168) |
| **Defending phase** | RR | DET | Lmax | L | ENTR | LAM | TT |
| Tackles performed | -.10 (p =.617) | **-.52 (p =.004)** | -.18 (p =.368) | **-.43 (p =.022)** | **-.45 (p =.016)** | -.27 (p =.161) | -.22 (p =.250) |
| Opponent wrong passes | -.12 (p =.544) | **-.38 (p =.046)** | -.24 (p =.224) | -.29 (p =.130) | -.31 (p =.111) | -.19 (p =.344) | -.06 (p =.768) |
| Shots to goal suffered | **.62 (p =.000)** | -.07 (p =.708) | .03 (p =.865) | -.08 (p =.674) | -.09 (p =.664) | -.15 (p =.441) | -.12 (p =.546) |
| Goals suffered | .2 (p =.300) | -.16 (p =.413) | -.07 (p =.71) | -.16 (p =.422) | -.16 (p =.429) | -.13 (p =.501) | -.16 (p =.403) |

From recurrence quantitative analysis, outcomes about the variability and recurrence of shape patterns were provided with seven different measures. In Fig 3, it is possible to visualize the distribution of the data for every measure considering different phases of the match. Because for the whole match, there is no ball possession condition between frames $i$ and $j$ beyond the distance threshold, it is expected that these recurrence plots would provide higher frequencies of recurrence states. On the other hand, during the attacking and defending phases, both distance threshold and ball possession condition needed to be satisfied to a recurrence state was considered between the pairs of $i$ and $j$. Although this was not the focus of the present study, from violin plots it seems that recurrence states are more present during the attacking phase when compared to the defending phase. Furthermore, concentrated data on the ends of the violin and not only around the mean/median value shows that teams have different behaviors in terms of presenting structural repetitive patterns when attacking and defending. Interestingly, recurrence plots were sensitive to show these differences not only among the teams but in specific periods of the match, as exemplified in Fig 2.

The correlation results showed that there is a clear association between recurrence state measures and performance indicators. Considering both the whole match and the attacking phase, there is a positive relationship between correct passes and recurrence rate. Interestingly, this outcome does not corroborate the ones reported by Lames and colleagues [24] in which no significant association was found between passes and recurrence rate of the average distance of all the players between each pair of timestamps, showing that positional variability and team structural organization do not reflect equally on the passing performance. Furthermore, the present study also reports a significant positive association between recurrence rate and shots to goal performed (during the attacking phase). In other words, teams that presented higher frequencies of correct passes and shots to goal also presented more recurrent states. The recurrence rate provides a measure of how often the time series revisits similar states or patterns (considering a specified distance threshold $\varepsilon$). A higher recurrence rate implies a

more regular behavior in the time series, while a lower recurrence rate indicates more irregular or chaotic dynamics [23]. This outcome refutes our initial hypothesis that the more variable the structure of the team shape is during ball possession, the higher the success in attacking actions. Our hypothesis was defined based on previous studies that reported teams and players that break the natural synchrony (i.e., the inter-teams' coordination and dyads coordination) with the opponents are more likely to have shots to goal at the end of attacking sequences [11, 36]. Therefore, from the divergences between the studies that analyzed different tactical variables, it is clear the importance of analyzing dynamical features regarding both micro (players) to macro (teams) structures and the role of the variability for success.

Although recurrence rate is positively correlated to correct passes performed and shots to goal, determinism was negatively correlated to goals scored and positively associated with ball possession when the whole match recurrence plots were analyzed, which implies that goals are unique events and something that has not happened previously is necessary to score [24]. Indeed, goals scored were also negatively correlated to $L_{max}$, entropy, and trapping time. A higher determinism value indicates a higher density of diagonal lines, which means the time series contains more deterministic and predictable patterns [23]. Both entropy and average diagonal line length are also related to diagonal line measures. Longer diagonal lines reflect that the time series tends to repeat certain patterns with relatively stable time intervals. A similar approach is visualized with the trapping time measure, which accounts for the time duration that a time series spends within a certain region of the recurrence plot, providing insights into the persistence or recurrence of the system's behavior. On the other hand, it is also important to emphasize that, when the attacking phase is taken into consideration, determinism is positively associated with correct passes performed and negatively correlated to wrong passes. Additionally, wrong passes were also negatively correlated to average diagonal line length, entropy, and trapping time. Thus, it seems that presenting a lower degree of determinism and persistent behaviour is important to football teams, however, they increase the chances of a higher number of wrong passes. In this sense, higher variability may represent the capability of a given team to constantly adapt its structure and present some level of unpredictability. On the other hand, it may represent the challenges that a given team has to present and maintain a structural pattern. Therefore, in the case of structural organization during attacks, variability is not inherently bad or good but there is a risk-reward trade that should be evaluated [37].

Very interesting associations between the recurrence plot measures and technical performance during defensive actions were found. The recurrence rate was positively correlated to shots to goal suffered. Therefore, presenting a rigid structure (i.e., with lower variability) during the defending phase may represent an increase in the opponent's odds of performing a shot to the goal. The negative correlations between tackles performed and determinism, average diagonal line length, and entropy corroborate the idea that defensive success is associated with higher variation and less deterministic behaviour regarding the tactical structures on the pitch. Additionally, the negative correlation of the determinism with opponent wrong passes suggests that more consistent patterns during the defending phase may facilitate the passing actions of the opponents. Again, this outcome refuses our initial hypothesis that a more structured shape would be associated with success in defensive actions. This hypothesis was based on the idea that modern football has a clear structure with the defending lines formed mainly by the defenders and midfielders. While it is positive to present a compact structure while defending [4], it is also important to present a regional flexible structure able to react to the opponent's actions. This is in accordance with a recent study [38] which reported the relevance of compact structure in areas close to the ball defensive success, but not for the entire team.

The proper analysis and interpretation of the recurrence plots and the quantitative measures extracted from them yield several considerable practical implications. Firstly, we

emphasize the advantages of a bidimensional representation of the behavioural pattern of a system, represented by the team's shape, throughout the match. In other words, the recurrence plot, with a single representation, allows the identification of clear patterns of structures of the teams' tactical organization on the pitch. Second, the different measures representing recurrence plot features are associated with performance indicators during attacking and defending actions. Therefore, having tracking and event data in hand, fast feedback for coaches and players may be available, with qualitative and quantitative analysis of the consistency of the team organization on the pitch. For example, there is a possibility of quantifying the recurrence states of the tactical organization during the first half of the match to present the coaches and players during interval time. This feedback would explain to staff and players how variable and unpredictable (or how 'rigid') their tactical organization was during the match. Also, such evaluation may guide training sessions regarding tactical actions. As it was proposed in a recent review [5], reference values of surface area can help technical staff create drills to enhance players' compactness while defending. For example, Rico-González and colleagues suggest exercises in which the pitch is subdivided into zones, and the defending team is penalized if the players occupy too many sub-zones. Considering the results of this study that present a rigid structure during the defending phase may lead to an increase in the opponent's odds of performing a shot to goal, technical staff may 'force' the defending players to adopt a different shape on the pitch for every pass performed by the opponent. For instance, every time that a pass is performed by the opponents, the defending players closest to the ball may pressure the attacking team reducing the distance to attackers. Rico-González and colleagues also suggest that it is positive to occupy a bigger area when attacking, so the team that has possession could be penalised if it occupies few sub-spaces. Because the present study showed that teams that presented higher frequencies of correct passes and shots to goal also presented more recurrent states (i.e., rigid structure), the technical staff may penalise the attacking team if the whole group does not move and occupy the neighbor zones together. These practical examples show the importance of both surface area magnitude and shape to enhance performance in football.

The present study has some limitations to be addressed in future research. Even considering the heterogeneity of the teams with different playing strategies, the outcomes reported may represent only the dynamics of Brazilian teams, so the extrapolation of our results to championships of other nationalities must be done with caution. It is important to emphasize that the key purpose of the present study was to explore the potential of dynamical systems measures applied to tactical analysis in football. Exploring the association between the present measures and performance indicators in larger samples is recommended. Finally, although the technical performance indicators presented provide valuable information about team performance and can discriminate between successful and unsuccessful teams, they did not represent, exclusively, the team performance. In other words, isolated analysis of correct or wrong passes, for instance, may not be representative of a team's performance considering the possibility of a given team making more mistakes and winning the match. There are complex factors that may lead to correct and wrong actions. Therefore, additional data about team performance, associated with contextual elements is recommended.

In conclusion, the key results showed that recurrence plots are promising tools to evaluate structural changes in the tactical organization of football teams throughout the match. It was possible to identify features of recurrence states of teams' tactics during defending and attacking phases, as well as of a general representation of the whole match. Results allowed to conclude that the success during attacking and defending actions, represented by traditional performance indicators, are closely associated with recurrent states and variability of players' organization on the pitch. Our findings reinforce the practical implication of recurrence plot as a valuable tool to represent teams' dynamics in football matches.

## Supporting information

**S1 Appendix. Recurrence quantitative analysis results.**
(ZIP)

## Author Contributions

**Conceptualization:** Felipe Arruda Moura, Murilo José de Oliveira Bueno, Fabio Giuliano Caetano, Sergio Augusto Cunha, Ricardo da Silva Torres.

**Data curation:** Felipe Arruda Moura, Murilo José de Oliveira Bueno, Fabio Giuliano Caetano, Maisa Silva, Ricardo da Silva Torres.

**Formal analysis:** Felipe Arruda Moura, Murilo José de Oliveira Bueno, Fabio Giuliano Caetano, Maisa Silva, Sergio Augusto Cunha, Ricardo da Silva Torres.

**Funding acquisition:** Felipe Arruda Moura.

**Investigation:** Felipe Arruda Moura, Murilo José de Oliveira Bueno, Fabio Giuliano Caetano, Ricardo da Silva Torres.

**Methodology:** Felipe Arruda Moura, Murilo José de Oliveira Bueno, Fabio Giuliano Caetano, Maisa Silva, Sergio Augusto Cunha, Ricardo da Silva Torres.

**Project administration:** Felipe Arruda Moura, Ricardo da Silva Torres.

**Resources:** Felipe Arruda Moura.

**Supervision:** Ricardo da Silva Torres.

**Validation:** Felipe Arruda Moura, Sergio Augusto Cunha.

**Visualization:** Felipe Arruda Moura.

**Writing – original draft:** Felipe Arruda Moura.

**Writing – review & editing:** Felipe Arruda Moura, Murilo José de Oliveira Bueno, Fabio Giuliano Caetano, Maisa Silva, Sergio Augusto Cunha, Ricardo da Silva Torres.

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
