## [Decision Letter · Decision Letter 0]

16 Apr 2024

PONE-D-23-37093Exploring the recurrent states of football teams' tactical organization on the pitchPLOS ONE

Dear Dr. Moura,

Thank you for submitting your manuscript to PLOS ONE. After careful consideration, we feel that it has merit but does not fully meet PLOS ONE’s publication criteria as it currently stands. Therefore, we invite you to submit a revised version of the manuscript that addresses the points raised during the review process.

We look forward to receiving your revised manuscript.

Kind regards,

Ersan Arslan, Ph.D.

Academic Editor

PLOS ONE

 [This work was supported by the Brazilian National Council for Scientific and Technological Development (CNPq) under grants \\#401004/2022-8, \\#200290/2022-3, and \\#305997/2022-0.].  

3.Thank you for stating the following in the Acknowledgments Section of your manuscript: 

[This work was supported by the Brazilian National Council for Scientific and Technological Development (CNPq) under grants \\#401004/2022-8, \\#200290/2022-3, and \\#305997/2022-0.].

  [This work was supported by the Brazilian National Council for Scientific and Technological Development (CNPq) under grants \\#401004/2022-8, \\#200290/2022-3, and \\#305997/2022-0.].  

Reviewers' comments:

Reviewer's Responses to Questions

**Comments to the Author**

1. Is the manuscript technically sound, and do the data support the conclusions?

Reviewer #1: Yes

Reviewer #2: Yes

2. Has the statistical analysis been performed appropriately and rigorously? 

Reviewer #1: Yes

Reviewer #2: Yes

3. Have the authors made all data underlying the findings in their manuscript fully available?

Reviewer #1: Yes

Reviewer #2: Yes

4. Is the manuscript presented in an intelligible fashion and written in standard English?

Reviewer #1: Yes

Reviewer #2: Yes

5. Review Comments to the Author

Reviewer #1: Overall, the article is well written and the topic is interesting. The authors need to make major changes before considering for publication.

Title: Exploring the recurrent states of football teams’ tactical organization on the pitch. Please, modify the title. In my opinion, you are generalising the results of this paper to a broader audience while you just have a sample of 28 Brazilian teams.

Introduction

L68: please, replace soccer by football. This is the term you have used throughout the manuscript.

L12-14: Please, add a citation to this sentence. Also, I suggest taking a look at a systematic review which was recently published and related to your study (https://pubmed.ncbi.nlm.nih.gov/35173369/) for your introduction

L26-L28: You are introducing the concept of variability here and this term is very important. Can you please add any information explaining what you mean by variability? If you decide to introduce concepts like entropy, please make sure that everyone understands what you are talking about.

L46, L54: Please, remove this abbreviation (RPs). You only mention it twice.

L47: Please, remove this abbreviation (RQA). You only mention it once.

L62: Explain what you mean by technical performance. This is a very important topic in your study.

L67: Please, avoid the recurrent use of “we” when writing a scientific document. Please, correct throughout.

L67-L73: In my opinion, this information is part of the methods. You may add a section in the methods section in which you talk about the study design or experimental approach to the problem.

L78: Based on my previous comment, I think that you may leave only the information related to the participants and remove the “data collection” as this could be part of the study design.

L81: data provided by (8)? Please, explain.

L81: How were the data collected?

L86: Please, correct the mistake (the the).

L103: Remove the abbreviation MFD from here and from the equation.

L111: defined as the following equation. Please, correct.

L175, L177: I suggest adding “technical” to …performance indicators…

L227: you talk about performance indicators here but this could be understood as tactical as well. I guess it would be “technical performance indicators”

L364-L365: Please, expand the practical applications section. This is a poor way to end such an interesting article. You are basically saying what we may already know. Please, think about how your results can have an impact on the field.

References: I found a few mistakes (e.g., avoid mixing uppercase/lowercase in all words, if you choose to add DOI, add all of them when it is possible, etc.) so please, review and follow journal’s guidelines.

Reviewer #2: The main focus of the present study was to explore the variability of surface area shapes of football teams on the pitch from recurrence plots and recurrence quantitative analysis, and the association with the performance indicators during the matches,' presents a well-written and innovative investigation. The literature background provides a strong foundation, and the application of recurrence analysis to football matches is a fresh approach.

However, to strengthen the paper for publication, a few areas require further clarification. The methods section would benefit from more technical details, and the discussion could be expanded on how these findings translate into actionable insights for coaches and analysts.

Indroduction

The indroduction section is well written. No comments to add.

Methods

1. L79: Concerning the 14 matches, did you apply any criteria in order to choose them? (red cards for example)

2. L80: The 366 players were outfield players? Have you excluded goalkeepers?

3. L85: Please provide the Ethics committee number

4. L88-90: Its not clear if the The DVideo software was used manually or automatically identified the actions.

5. L93-44: If there was manual registration (actions, success/no success and responsible player) why you provided the inter- and intra-rater agreement for the system and you didn't conduct new reliability tests?

6. L168-170: You should provide this information in a different paragraph titled "Technical Performance Indicators" or something similar. Its irrelevant to the statistical analysis. Moreover, the first sentence "The technical performance during attacking and defending phases were defined by the nine indicators" needs clarification. You need to provide definitions for the indicators chosen. How you define the "wrong pass", "correct pass", "tackles performed" etc

7. L169: Replace "right pass" with "correct pass" throughout the manuscript.

8. L177: Why you included the "opponent wrong pass"indicator to defending phase? There could be numerous reasons for the wrong pass, that don't concern the behavior of the defending team.

Results

The results section is well presented. No comments to add.

Discussion

1.L274-275: The attacking phase includes right passes. I think this sentence needs re-writing.

2.L345-348: Is it possible at the moment to have this fast feedback live during matches at the bench? I believe the last word “performance” should be modified because its an overwhelming statement. You took into account specific performance indicators that maybe are not representative of the performance as a “whole”. Moreover, you should try to enrich your practical implication section, in order to provide coaches and analysts specific “take-home” messages that I am afraid that are not present in this version.

3.L349-356: Based on the comments before concerning Methods section, you should add more limitations.

4.L364-365: Again be careful with your statement considering recurrence plot as a tool during football matches. Can you use it live?

6. PLOS authors have the option to publish the peer review history of their article (what does this mean?). If published, this will include your full peer review and any attached files.

Reviewer #1: No

Reviewer #2: No

---

## [Author Response · Author response to Decision Letter 0]

21 May 2024

Response to Reviewers

According to the Editor's recommendations, we are resubmitting the paper “Exploring the recurrent states of football teams' tactical organization on the pitch (New title: Exploring the recurrent states of football teams' tactical organization on the pitch during Brazilian official matches”). We would like to thank the editor and the reviewers for their suggestions, which have contributed positively to the quality of the study.

All the suggestions pointed out by the reviewers were accepted and we highlighted all changes on our manuscript in coloured text. We have also included point-by-point responses to the reviewers’ comments.

 ASSOCIATE EDITOR’S COMMENTS 

1 Thank you for submitting your manuscript to PLOS ONE. After careful consideration, we feel that it has merit but does not fully meet PLOS ONE’s publication criteria as it currently stands. Therefore, we invite you to submit a revised version of the manuscript that addresses the points raised during the review process.

Thank you for stating the following financial disclosure: 

 [This work was supported by the Brazilian National Council for Scientific and Technological Development (CNPq) under grants \\#401004/2022-8, \\#200290/2022-3, and \\#305997/2022-0.]. 

Thank you for stating the following in the Acknowledgments Section of your manuscript: 

[This work was supported by the Brazilian National Council for Scientific and Technological Development (CNPq) under grants \\#401004/2022-8, \\#200290/2022-3, and \\#305997/2022-0.].

 [This work was supported by the Brazilian National Council for Scientific and Technological Development (CNPq) under grants \\#401004/2022-8, \\#200290/2022-3, and \\#305997/2022-0.]. 

Please include your full ethics statement in the ‘Methods’ section of your manuscript file. In your statement, please include the full name of the IRB or ethics committee who approved or waived your study, as well as whether or not you obtained informed written or verbal consent. If consent was waived for your study, please include this information in your statement as well.

We would like to thank the editor for the suggestions and for the opportunity to address all the comments and questions. 

In the current version of the manuscript, we carefully addressed all raised issues. Point-by-point responses are provided below, and associated modifications are highlighted in the document.

- We have edited the manuscript accordantly with PLOS ONE’s style requirements.

- All the requirements concerning the financial disclosure were addressed. 

- Ethics Committee information were included.

 Reviewer 1 - COMMENTS 

1 Overall, the article is well written and the topic is interesting. The authors need to make major changes before considering for publication.

Title: Exploring the recurrent states of football teams’ tactical organization on the pitch. Please, modify the title. In my opinion, you are generalising the results of this paper to a broader audience while you just have a sample of 28 Brazilian teams. 

We appreciate the reviewer’s suggestions. All the suggestions were taken into consideration.

-----

Thank you for the suggestion about the title. Our intention with the terms “Exploring the recurrent states” was to avoid generalisation and to emphasise the relevance of the recurrence plot as a method to investigate the dynamical properties of tactical-based time series. However, we understand your concern about generalising the results. Therefore, we emphasise that the study was developed with Brazilian matches. Thus, the new title is: ‘Exploring the recurrent states of football teams' tactical organization on the pitch during Brazilian official matches’. 

2 Introduction

L68: please, replace soccer by football. This is the term you have used throughout the manuscript. Thanks. The term was replaced accordingly.

3 L12-14: Please, add a citation to this sentence. Also, I suggest taking a look at a systematic review which was recently published and related to your study (https://pubmed.ncbi.nlm.nih.gov/35173369/) for your introduction Thank you for the suggestion. A reference was inserted for L12-14. 

Thank you for the indication of the systematic review. We added this reference in the introduction and discussion sections. 

4 L26-L28: You are introducing the concept of variability here and this term is very important. Can you please add any information explaining what you mean by variability? If you decide to introduce concepts like entropy, please make sure that everyone understands what you are talking about. Thank you for this suggestion. We have included a definition of variability analysis, highlighting some methods traditionally investigated in sport analytics, including entropy. 

5 L46, L54: Please, remove this abbreviation (RPs). You only mention it twice. The abbreviation was removed.

6 L47: Please, remove this abbreviation (RQA). You only mention it once. The abbreviation was removed.

7 L62: Explain what you mean by technical performance. This is a very important topic in your study. We thank the reviewer for the suggestion. In this case, we represented technical performance by traditional game-related statistics as passes performed, shots to goal, goals scored, etc. We included this information when we first mentioned ‘technical performance’.

8 L67: Please, avoid the recurrent use of “we” when writing a scientific document. Please, correct throughout. We removed most of the first-person voice in the manuscript, remaining clear sentences in self-reference, following the APA manual which recommends the use of first person in scientific writing.

9 L67-L73: In my opinion, this information is part of the methods. You may add a section in the methods section in which you talk about the study design or experimental approach to the problem. This paragraph was reformulated, and the information was concentrated on the methods section, in a subsection for study design. Thank you for this suggestion.

10 L78: Based on my previous comment, I think that you may leave only the information related to the participants and remove the “data collection” as this could be part of the study design. Based on your suggestion, we presented general information about study design and removed “data collection” from the subsection head.

11 L81: data provided by (8)? Please, explain. Sorry about this confusing sentence. We intend to explain that we used the same data of the study numbered as 8. The sentences of this section were reformulated. Thank you. 

12 L81: How were the data collected? Additional information was included about DVideo software and data collection.

13 L86: Please, correct the mistake (the the) Done. Thank you.

14 L103: Remove the abbreviation MFD from here and from the equation. The abbreviation was removed.

15 L111: defined as the following equation. Please, correct. The sentence was corrected.

16 L175, L177: I suggest adding “technical” to …performance indicators… The term was added to the suggested sentences.

17 L227: you talk about performance indicators here but this could be understood as tactical as well. I guess it would be “technical performance indicators” Agreed. The term “technical” was added in the sentence.

18 L364-L365: Please, expand the practical applications section. This is a poor way to end such an interesting article. You are basically saying what we may already know. Please, think about how your results can have an impact on the field. Thank you for this suggestion. We have included some clear practical applications, combining with the review that you suggested for the introduction.

19 References: I found a few mistakes (e.g., avoid mixing uppercase/lowercase in all words, if you choose to add DOI, add all of them when it is possible, etc.) so please, review and follow journal’s guidelines. All the references were updated. Thank you for the detailed check.

Reviewer 2 - COMMENTS 

The main focus of the present study was to explore the variability of surface area shapes of football teams on the pitch from recurrence plots and recurrence quantitative analysis, and the association with the performance indicators during the matches,' presents a well-written and innovative investigation. The literature background provides a strong foundation, and the application of recurrence analysis to football matches is a fresh approach.

However, to strengthen the paper for publication, a few areas require further clarification. The methods section would benefit from more technical details, and the discussion could be expanded on how these findings translate into actionable insights for coaches and analysts. 

We appreciate the reviewer’s suggestions and detailed reading of our manuscript. All the suggestions were taken into consideration, especially the methods and discussion sections. 

-----

Introduction

The introduction section is well written. No comments to add. We appreciate the comments.

Methods

1. L79: Concerning the 14 matches, did you apply any criteria in order to choose them? (red cards for example) Thank you for concern. The matches were selected by convenience, considering that the datasets were provided for us from previous studies. All the 14 matches are complete and with no red cards or injuries that may have resulted in longer periods with less than 10 outfield players for each team. We have included this information in the methods section. 

L80: The 366 players were outfield players? Have you excluded goalkeepers? No, we did not exclude the goalkeepers from tracking but from the surface area representation only.

L85: Please provide the Ethics committee number We have included the Ethics committee number

L88-90: Its not clear if the The DVideo software was used manually or automatically identified the actions. The DVideo software was used to manual register the technical actions of the players during the football matches. We highlighted this information in the manuscript.

L93-44: If there was manual registration (actions, success/no success and responsible player) why you provided the inter- and intra-rater agreement for the system and you didn't conduct new reliability tests? We agree with the reviewer. Indeed, we evaluated the reliability for the manual register procedure, i.e., the inter and intra-rater agreement. The sentence was reformulated. Thank you very much. 

L168-170: You should provide this information in a different paragraph titled "Technical Performance Indicators" or something similar. Its irrelevant to the statistical analysis. Moreover, the first sentence "The technical performance during attacking and defending phases were defined by the nine indicators" needs clarification. You need to provide definitions for the indicators chosen. How you define the "wrong pass", "correct pass", "tackles performed" etc We have included a new subsection for the technical performance indicators, and we have presented their definitions.

L169: Replace "right pass" with "correct pass" throughout the manuscript. Done. Thank you.

L177: Why you included the "opponent wrong pass"indicator to defending phase? There could be numerous reasons for the wrong pass, that don't concern the behavior of the defending team. We partially agree with the reviewer. Although we understand that several reasons may lead to a wrong pass, the same interpretation may be made for all technical action. In other words, even correct actions, such as passes, shots on goal, are dependent on several circumstances that include the teammates and opponent players. However, we associated each indicator to the principles that guide the team action while attacking and defending. Considering that the defending principles are to regain ball possession or to lead the opponent to perform a mistake, we understand that opponent wrong passes were associated with a defending phase, i.e., the defending team has success in its principle. However, we completely understand your concern about the generalisation of this indicator, and we have added this point to the limitations of the study. 

Results

The results section is well presented. No comments to add. We appreciate the comments.

Discussion

1.L274-275: The attacking phase includes right passes. I think this sentence needs re-writing. This sentence was reformulated. Thank you for the detailed review.

L345-348: Is it possible at the moment to have this fast feedback live during matches at the bench? I believe the last word “performance” should be modified because its an overwhelming statement. You took into account specific performance indicators that maybe are not representative of the performance as a “whole”. Moreover, you should try to enrich your practical implication section, in order to provide coaches and analysts specific “take-home” messages that I am afraid that are not present in this version. Thank you for this interesting question. Nowadays, with recent advances of player tracking, it would be possible to provide live feedback to coaches during the match. However, that was not our intention. We mentioned fast feedback as a possibility of to quantify the recurrence states of the tactical organization during the first half of the match, for example, to present the coaches and players during interval time. This feedback would explain to staff and players how variable and unpredictable, or how ‘rigid’ their tactical organization was during the match. We reformulated these sentences. 

We also addressed your suggestion (and reviewer´s 1 suggestion) about to enrich the practical implications. We presented some examples for training sessions based on the results presented. Thank you. 

L349-356: Based on the comments before concerning Methods section, you should add more limitations. We have included the limitation about the technical performance indicators and that the outcome may be interpreted with caution. 

L364-365: Again be careful with your statement considering recurrence plot as a tool during football matches. Can you use it live? Nowadays, with recent advances of player tracking, it would be possible to provide live feedback to coaches during the match. However, we have decided to remove considering that specific championships may restrict the access to such information during the match. Thank you.

---

## [Decision Letter · Decision Letter 1]

23 Jul 2024

Exploring the recurrent states of football teams' tactical organization on the pitch during Brazilian official matches

PONE-D-23-37093R1

Dear Dr. Torres,

We’re pleased to inform you that your manuscript has been judged scientifically suitable for publication and will be formally accepted for publication once it meets all outstanding technical requirements.

Kind regards,

Haroldo V. Ribeiro

Academic Editor

PLOS ONE

Additional Editor Comments (optional):

The original editor of your manuscript was no longer available, so I have been assigned to step in. I observed that two experts had previously reviewed your manuscript and offered a series of suggestions, which you have addressed in your revised submission. This revised version was returned to these two experts; however, only one responded, recommending publication as is. I then conducted my own review of your manuscript and your responses to the comments from the initial submission. My assessment aligns with that of both reviewers. I commend you on your excellent work and am pleased to accept your manuscript in its current form.

Reviewers' comments:

Reviewer's Responses to Questions

**Comments to the Author**

1. If the authors have adequately addressed your comments raised in a previous round of review and you feel that this manuscript is now acceptable for publication, you may indicate that here to bypass the “Comments to the Author” section, enter your conflict of interest statement in the “Confidential to Editor” section, and submit your "Accept" recommendation.

Reviewer #2: All comments have been addressed

2. Is the manuscript technically sound, and do the data support the conclusions?

Reviewer #2: Yes

3. Has the statistical analysis been performed appropriately and rigorously? 

Reviewer #2: Yes

4. Have the authors made all data underlying the findings in their manuscript fully available?

Reviewer #2: Yes

5. Is the manuscript presented in an intelligible fashion and written in standard English?

Reviewer #2: Yes

6. Review Comments to the Author

Reviewer #2: Based on the overall quality of the research and the authors' thorough response to the reviewers' comments, I recommend acceptance of this manuscript for publication in PLOS.

7. PLOS authors have the option to publish the peer review history of their article (what does this mean?). If published, this will include your full peer review and any attached files.

Reviewer #2: **Yes: **Vasilis Armatas

---

## [Editor Report · Acceptance letter]

1 Aug 2024

PONE-D-23-37093R1 

PLOS ONE

Dear Dr. Torres, 

I'm pleased to inform you that your manuscript has been deemed suitable for publication in PLOS ONE. Congratulations! Your manuscript is now being handed over to our production team.

Kind regards, 

on behalf of

Dr. Haroldo V. Ribeiro 

Academic Editor

PLOS ONE